# Disentangling the Functional Role of Fungi in Cold Seep Sediment

Erfan Shekarriz,[a,b,c,d] Jiawei Chen,[a,b,c,d] Zhimeng Xu,[a,b,c,d] (ID) Hongbin Liu[a,b,c,d]

[a]Southern Marine Science and Engineering Guangdong Laboratory (Guangzhou), Guangzhou, China
[b]Department of Ocean Science, Hong Kong University of Science and Technology, Hong Kong, China
[c]Department of Ocean Science, The Hong Kong University of Science and Technology, Hong Kong, China
[d]Hong Kong Branch of the Southern Marine Science and Engineering Guangdong Laboratory (Guangzhou), The Hong Kong University of Science and Technology, Hong Kong, China

**ABSTRACT** Cold seeps are biological oases of the deep sea fueled by methane, sulfates, nitrates, and other inorganic sources of energy. Chemolithoautotrophic bacteria and archaea dominate seep sediment, and their diversity and biogeochemical functions are well established. Fungi are likewise diverse, metabolically versatile, and known for their ability to capture and oxidize methane. Still, no study has ever explored the functional role of the mycobiota in the cold seep biome. To assess the complex role of fungi and fill in the gaps, we performed network analysis on 147 samples to disentangle fungal-prokaryotic interactions (fungal 18S and prokaryotic 16S) in the Haima cold seep region. We demonstrated that fungi are central species with high connectivity at the epicenter of prokaryotic networks, reduce their random-attack vulnerability by 60%, and enhance information transfer efficiency by 15%. We then scavenged a global metagenomic and metatranscriptomic data set from 10 cold seep regions for fungal genes of interest (hydrophobins, cytochrome P450s, and ligninolytic family of enzymes); this is the first study to report active transcription of 2,500+ fungal genes in the cold seep sediment. The genera *Fusarium* and *Moniliella* were of notable importance and directly correlated with high methane abundance in the sulfate-methane transition zone (SMTZ), likely due to their ability to degrade and solubilize methane and oils. Overall, our results highlight the essential yet overlooked contribution of fungi to cold seep biological networks and the role of fungi in regulating cold seep biogeochemistry.

**IMPORTANCE** The challenges we face when analyzing eukaryotic metagenomic and metatranscriptomic data sets have hindered our understanding of cold seep fungi and microbial eukaryotes. This fact does not make the mycobiota any less critical in mediating cold seep biogeochemistry. On the contrary, many fungal genera can oxidize and solubilize methane, produce methane, and play a unique role in nutrient recycling via saprotrophic enzymatic activity. In this study, we used network analysis to uncover key fungal-prokaryotic interactions that can mediate methane biogeochemistry and metagenomics and metatranscriptomics to report that fungi are transcriptionally active in the cold seep sediment. With concerns over rising methane levels and cold seeps being a pivotal source of global methane input, our holistic understanding of methane biogeochemistry with all domains of life is essential. We ultimately encourage scientists to utilize state-of-the-art tools and multifaceted approaches to uncover the role of microeukaryotic organisms in understudied systems.

**KEYWORDS** biogeochemistry, cold seep, filamentous fungi, fungi, metagenomics, metatranscriptomics, methanotrophs, network analysis, nonhuman microbiome, target amplicon sequencing

Address correspondence to Hongbin Liu, liuhb@ust.hk.

The authors declare no conflict of interest.

**F**ungi play pervasive and vital roles in nature, with unique functional characteristics that mediate entire ecosystems (1). Their unique functional characteristics allow them to persevere and dominate most environments as a second runner to bacteria and the most abundant eukaryote (2). While our fundamental knowledge of the mycobiota originates from the terrestrial ecosphere, recent explorations highlight their overlooked contribution to aquatic systems (3–6). The role of fungi in the deep sea has sparked particular scientific interest (7–10). We believe this fact to be based on three significant grounds: first, that fungi are abundant and diverse in critical deep-sea biomes (11, 12); second, that they produce valuable enzymes and low-molecular-weight compounds of interest (7) (some of which have already been extracted [13, 14]); and lastly, that the inherent versatility of fungal metabolism makes them favorable candidates for dominating the extreme deep-sea environments (15). Fungi have adaptive life cycles (16–19), taking advantage of both asexual reproduction for copy number dominance and asexual reproduction to enrich genetic diversity. They can readily acquire genes from neighboring bacteria via horizontal gene transfer (20) and have evolutionarily distinct proteins that break down high-molecular-weight carbon compounds (e.g., ligninolytic enzymes) that often undergo insertion and deletion events (21, 22). All features mentioned above facilitate both intergenerational and cross-generational metabolic adaptation.

With all being said, our understanding of the precise functional role of fungi in the deep sea is limited. Although the high diversity of methane gas cold seep fungi was first reported in 2006 (23), to our knowledge, at the time of writing this paper only six other reports have explicitly followed up on the phylogeny and abundance of fungi in cold seep sediment (24–29). All findings are consistent in showing that cold seep fungi are diverse and abundant (with one study exploring antibacterial properties), yet the studies failed to identify any direct molecular evidence of their functional or metabolic role (relying on highly extrapolated taxon-specific assumptions). All the studies likewise utilized genomic data and did not assess whether fungi are transcriptionally active in cold seep sediment. The scarcity of functionally driven fungal cold seep studies we assume to be due to the historic insufficiency of scientific technology that enables us to access, sample, extract, analyze, and ultimately understand fungi and their roles in harsh deep-sea environments (30, 31). Even more, it could be due to the challenges we face when reconstructing and annotating eukaryotic genomes using metagenomic approaches in an omics-oriented world (32, 33). The facts do not take away from the broad metabolic adaptations of fungi and their unfound ecological potential in the seep environment.

Cold seeps are biological oases of the nutrient-deprived deep sea, driven by methane gas, sulfates, nitrates, phosphates, and other hydrocarbon-rich compounds and oils (34). Chemolithoautotrophic bacteria and archaea dominate the microbial ecosystem and use the aforementioned inorganic energy sources to drive higher trophic-level diversity (35). Three domains of mycology research relevant to cold seep sediment have inspired our hypothesis that fungi play dominating roles in biogeochemical cycling. The first is the ability of fungi to solubilize methane for methanotrophic bacterial degradation (36) carried out by hydrophobins, a family of amphipathic surface proteins secreted by filamentous fungi that solubilize hydrophobic compounds (e.g., methane, toluene, and benzene) by increasing their partition coefficient in water (12, 37, 38). The second concerns fungal hydrocarbon compound degradation and bioremediation. Yeast forms of fungi were first seen to degrade methane in 1979 (39). Fungal genomes that carry and express the cytochrome P450 gene family (40) can oxidize ethane, toluene, polycyclic aromatic molecules, and a wide variety of hydrocarbons owing to P450s' well-studied substrate heterogeneity and versatility. Under the right conditions, fungal P450s degrade methane (41). The ability to solubilize and oxidize methane is essential in cold seep environments, as methane is a highly insoluble gas protected by a layer of water (42, 43) and hence not readily available for uptake by methanotrophic archaea and bacteria under Henry's law (44). The third and last field of interest is the composition of cold seep sediment, mainly constituted by 68% to 82% lignin (45); lignins are a high-molecular-weight family of compounds resulting from the death of megafauna on seep sediment and almost exclusively degraded by fungi in

other soil environments (46). All traits mentioned above, combined with the wide variety of adaptations that fungi employ to survive in harsh conditions, make them suitable, reasonable, and valuable candidates for the cold seep ecosystem.

While fungi are likely important players in cold seep microbiomes, studying their functional role in remote and difficult-to-access environments is a nontrivial task. Hence, to assess and ascertain the functional input of the mycobiota to the environment, we employed a multiomics approach that integrates community composition data, network interaction analysis, and metaomics sequencing (metagenomics and metatranscriptomics) of the cold seep surface sediment (presumably the most essential biogeochemically active layer that mediates methane export) (47, 48). Our data set is large ($n = 147$), sampling three local seep regions in the Haima cold seep as well as a neighboring control. It is comprehensive in that it measured numerous environmental factors associated with sampling clusters. Lastly, it is representative in that it included both amplicon community composition analysis for our local samples and metaomics data from a global data set of 10 seep sites (49). While it is challenging to infer fungal ecosystem roles from only genomic data (since fungi also exhibit behavioral characteristics), interdisciplinary advances enable scientists to assess the holistic impact and importance of fungi on an ecosystem level (50). Network analysis (NA) derived from mathematical graph theory specifically allows us to do so by understanding biological relationships through co-occurrence analysis specialized for microbiome data set properties (51) and has proven to reveal meaningful and experimentally validated biotic interactions on a species level (52, 53) while also highlighting the complexity of diverse microbial interactions. Analysis of network topography and connectivity patterns allows scientists to model microbial community members' metabolic connections and assess their ability to stabilize and impact the microbiome as a whole.

Our study ultimately took advantage of and applied interdomain and environmental-data-conscious network inference to the cold seep microbiome with aim to understand the contribution of fungi to cold seep environments as a community while disentangling scientifically plausible bacterial-fungal interactions as hypotheses for future validation. Overall, we sought to unravel the complex biogeochemically driven ecosystem of cold seeps and the role of the fungi within using a robust and confident statistical and functional multiomics approach that leverages abiotic measurements (methane, sulfate, sulfide, dissolved inorganic carbon). To our knowledge, this is the first exploration of the functional role of the fungal community and multidomain network modeling in cold seep sediment.

## RESULTS

**Environmental context of cold seep and control samples.** Four different sampling clusters were chosen from the Haima cold seep (or neighboring nonseep) regions in the South China Sea (SCS) and analyzed for environmental parameters. We used both remotely operated videography and nutrient depth profiles to confirm the activities of three cold seep sites (ROV1, ROV2, and ROV3) and a benthic sediment sampling site (ROV5) as a control (Fig. 1). The seep surface layer sediment had detectable gaseous bubbling and high megafauna diversity with higher methane and sulfide concentrations across all depths. The depth profile was used to evaluate the extent of cold seep activity, with ROV1 and ROV2 marked as the most active and typical sulfate-methane transition zones (SMTZ) at the 10- to 13-cm depth interval. The SMTZ is the depth at which the highest combination of methane and sulfate exists, enhancing sulfate reduction-coupled anaerobic oxidation of methane by anaerobic methanotrophic archaea (ANME) and sulfate-reducing bacteria (SRB) (54). This makes the SMTZ the hub of the biogeochemical processing of methane and where we expect the most vital and dynamic interactions of fungi and other community members to occur. Despite a non-distinct SMTZ, we consider ROV3 a low-activity cold seep owing to its ability to bubble methane, host megafauna, and display higher concentrations of sulfides and methane than the inactive ROV5. The depth profiles of all measured metadata variables were also reported (see Fig. S1 and S2 in the supplemental material).

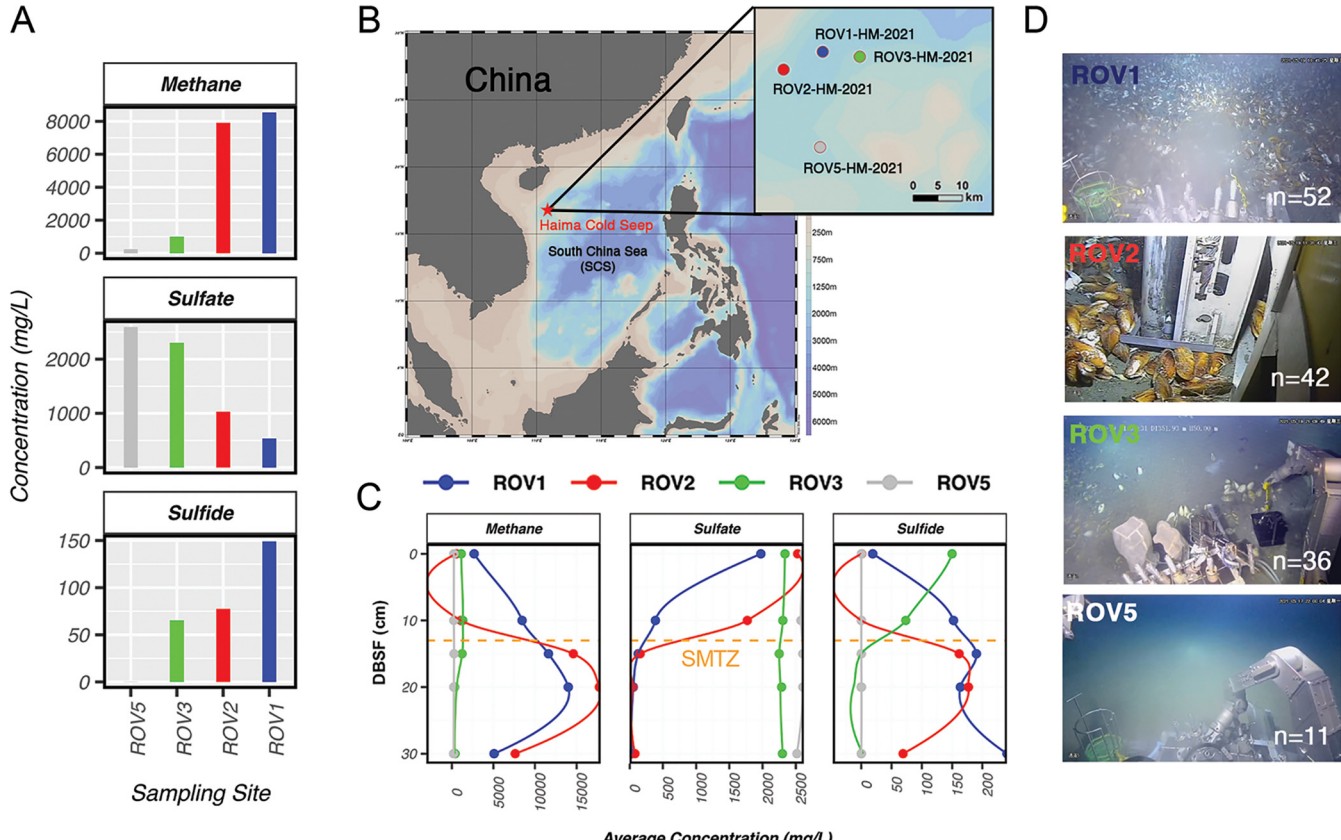

**FIG 1** Sampling site locations and abiotic measurements. (A) Average methane, sulfate, and sulfide concentrations of all samples measured. (B) Relative location of the Haima cold seep (ROV1, ROV2, ROV3) and neighboring control region (ROV5) sampling sites. (C) Vertical depth profile (depth below sea floor [DBSF]) of average methane, sulfate, and sulfide concentrations of all sediment samples measured. (D) Video snapshots of surface layer megafauna and bubbling for all sampling sites.

**Analysis of microbial communities and fungal diversity in cold seep sediment.** The microbial community (fungal 18S and prokaryotic 16S) compositions of cold seep surface sediment samples (ROV1, ROV2, and ROV3) were explored and compared to that of the control (ROV5). We analyzed the species richness of all samples using a 10,000-fold rarefaction procedure with the iNEXT package, where the coverage extrapolation is made through a robust bootstrapping statistical framework with a 95% confidence interval (55). We confirmed that all samples had full coverage of amplicon sequence variants (ASVs) (Fig. S3) in both 16S and fungal 18S analysis. Cold seep prokaryotic and fungal communities were more diverse in terms of richness than the control ROV5 (Fig. 1E). We also found no significant correlation between richness and abiotic factors (Fig. S4).

The phylum-level phylogenetic diversity was high in both control and seep environments in that we observed at least five representative ASVs from each clade (Fig. 1A). A proportion of fungi were left unidentified, likely due to the high sequence divergence of 18S genes from isolated environments and a lack of robust metagenome-assembled genomes (MAGs) of the eukaryotic lineage database entries (a standard caveat in deep-sea community studies) (56). The distribution of phylum-level abundances was skewed (Fig. S5), with Basidiomycota dominating all sites (50 to 69%), followed by Ascomycota (14 to 64%), two unidentified clades (3 to 8%), and the remainder of phyla in very low abundance (<1%). This contradicted previous studies showing that basal fungi (e.g., Mucoromycota and Glomeromycota) dominate the community (25). Fungi were the most abundant protists overall (57% ± 19%) (Fig. 2D), but Alveolata and, in some cases, Stramenopiles dominated the surface of ROV1 (Fig. S6). We presume this with the parasitic relations of unicellular eukaryotes with surface-layer megafauna in cold seep sites (Fig. 1) (57, 58).

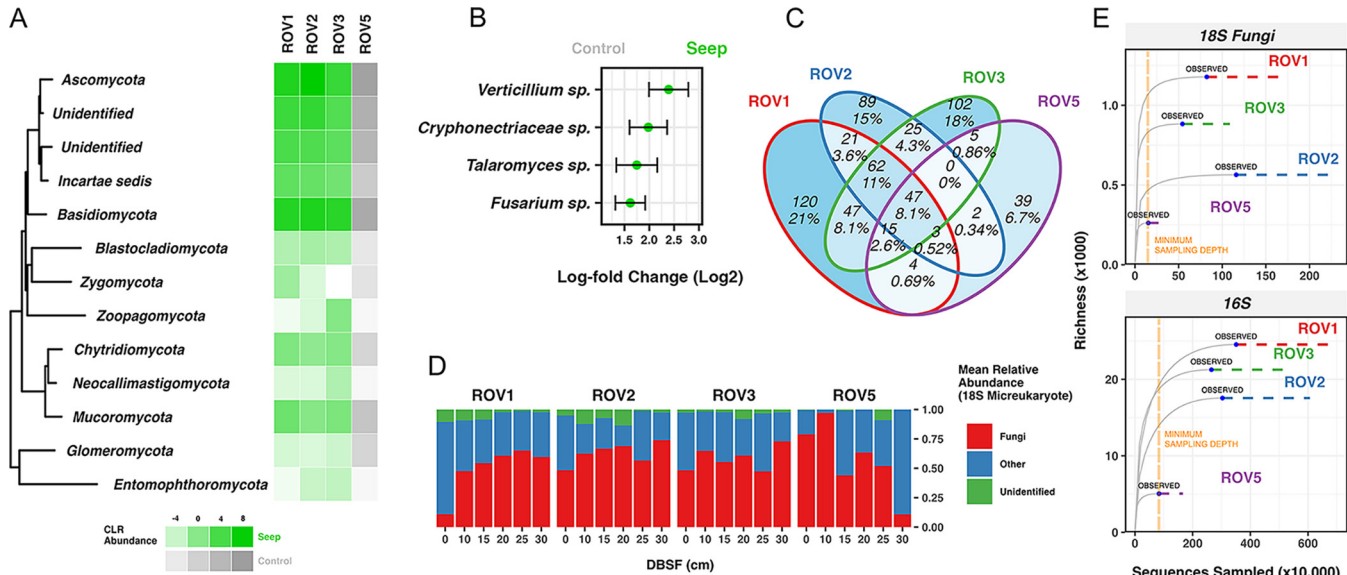

**FIG 2** Community composition analysis using 18S fungal and 16S prokaryotic primers. (A) Centered log ratio (CLR) phylum-level abundance heat map of different sampling sites (generated by 18S MAFFT alignment and plotted using ggtree packaged in R). (B) Differential abundance analysis using LinDA (adjusted $P$ value < 0.05) with fungal 18S ASVs labeled to genus level that best distinguished seep and control samples. (C) Venn diagram of number and percentage of shared 18S fungal ASVs by four different sampling sites. (D) Average relative abundance bar chart of fungal 18S rRNA gene region compared to all microeukaryotes in different depths and sites. (E) Site-merged rarefaction curves produced by the "iNEXT" framework using a 95% confidence interval extrapolation (dashed line) and 10,000 bootstrapping iterations, with average richness reported (gray line). Vertical dashed orange line represents the minimum sequencing depth and the point of comparison for richness between different sites.

All sites shared 47 fungal ASVs (control and seep), but overall cold seep samples (ROV1, ROV2, and ROV3) had a higher percentage of ASVs in common with each other (11%) than they individually had with ROV5 (<1%), which suggests that some fungal species have a niche preference for the cold seep biome. Differential abundance analysis was performed using the compositional LinDA framework (59) to identify precisely which group of taxa best separates the seep from control site samples. Four Ascomycetes ASVs were associated with seep sediment (*Fusarium oxysporum*, *Verticillium longisporum*, *Talaromyces purpureogenus*, and a *Cryphonectriaceae* sp.). At this point, we refrain from any taxonomy-based extrapolations of fungal function and explore this below. The Phyloseq file for accessing ASV tables and taxonomic profiles of 18S fungal and 16S sequences is available (see "Data and code availability" below).

**Fungi enhance robustness, information transfer, and connectivity of cold seep networks.** Inferring function from phylogeny and abundance of microbes alone can be misleading and challenging, as it has been shown repeatedly that low-count microbial members have primary ecological functions that drive the community (60). Fungi, on average, carry larger genomes relative to neighboring bacteria and archaea (22), with a complex set of metabolic strategies and behavioral adaptations (61) that give them dominant roles in microbial communities (even in low abundances). To study the complex role of fungi in a comprehensive manner, as well as their importance in cold seep communities, we implemented network analysis (Fig. 3) using the SpiecEasi sparse and low rank (SLR) method to account for the compositionality, zero inflation, and effect of latent as well as measured environmental factors on the false-positive and false-negative biological relationships (62, 63).

Network attack vulnerability ($V$) can mimic the stochastic removal of organisms from an environment and was implemented following the method of Iyer et al. (64) and modified to include 10,000 bootstrapping iterations to assess significance. We found that the inclusion of fungi overall reduced the vulnerability of biological networks to random attack ($P < 0.0001$) (Fig. 3A). The most robust network was the bacterial-archaeal-fungal (BAF) network ($V_{mean} = 0.152$), followed by the bacterial-fungal (BF) ($V_{mean} = 0.197$), bacterial (B) ($V_{mean} = 0.249$), and bacterial-archaeal (BA) ($V_{mean} = 0.248$) networks. Archaea did

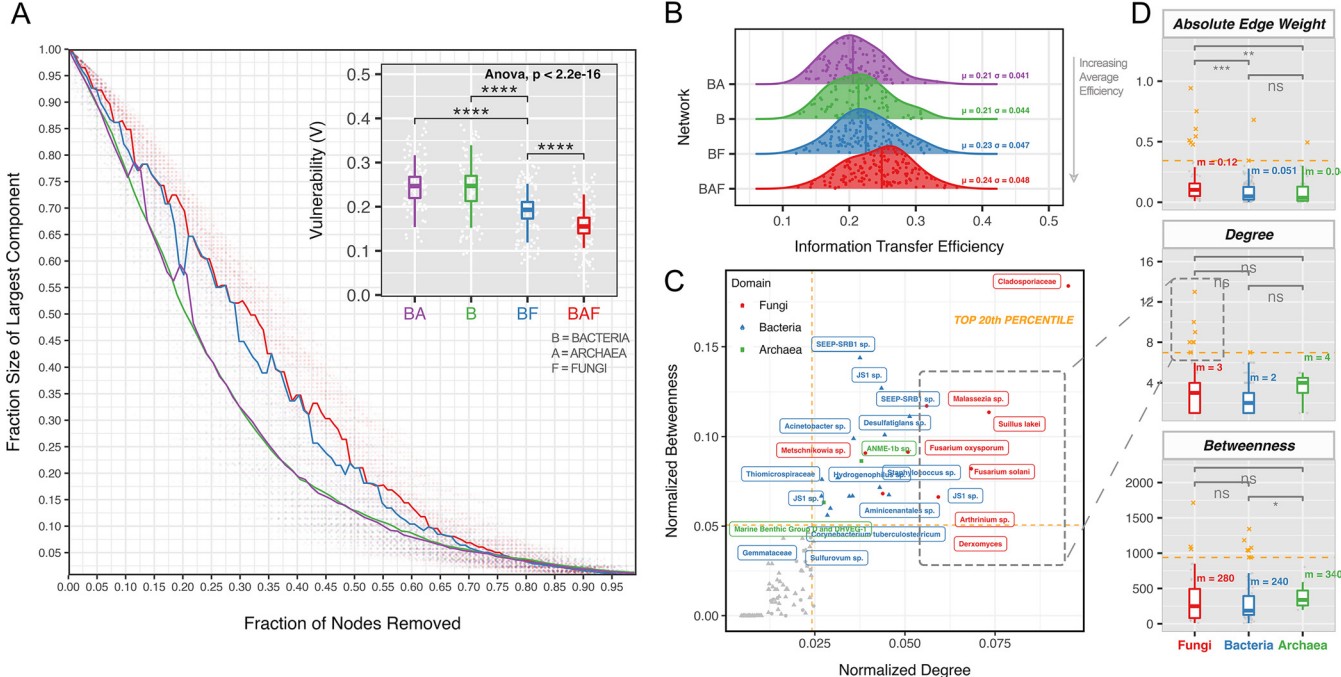

**FIG 3** Cold seep network analysis of robustness, efficiency, and connectivity of BAF, BA, BF, and Ba networks. (A) Random-attack robustness curves with 10,000 iterations, inferred vulnerability (V), and *P* value significance on the difference in vulnerability. (B) Efficiency distribution curves of networks in order of increasing average efficiency, with dots representing each node within a network. (C) Cold seep multidomain network (18S and 16S) keystone species analysis on the basis of degree and betweenness, with top 20th percentile of each axis colored and reported as keystone species. (D) Box plots of BAF network absolute edge weight, degree, and betweenness of its constituent archaea (green), bacteria (blue), and fungi (red), with *P* values of mean differences calculated using Wilcoxon signed-rank test. The top 20th percentile of each factor presented as crosses and colored in orange. *, $P < 0.05$; **, $P < 0.01$; ***, $P < 0.001$; ****, $P < 0.0001$. ns, not significant.

not significantly reduce the random-attack vulnerability of bacterial networks but did so when fungi were involved. Fungal presence in prokaryotic networks decreased vulnerability by 60%. While attack robustness gives an approximation of connectivity and community health, nodal efficiency measures the ability of a taxon/node to propagate information with other network members (65). It could potentially represent the efficiency by which chemical and behavioral signals (e.g., nutrients, secondary metabolites, and phagotrophy) are exchanged among members of the taxon. We performed this analysis because efficiency can, at times, be inversely correlated with robustness, as the inclusion of more connected edges between members of a taxon makes the network more resilient to attacks yet less efficient at creating the shortest path between two ASVs (66). Contradictory to this finding, the BAF network was most efficient ($\mu = 0.24$), followed by the BF network ($\mu = 0.23$) (Fig. 3B).

Knowing the robust and efficient nature of the BAF network, we evaluated the specific contributions of bacteria, archaea, and fungi to its structural properties. The averaged absolute edge weights to and from fungi were significantly higher than those of both archaea and bacteria (Fig. 3D); fungus-associated connections were either largely positive or negative. While the median connectivity (degree and number of connections to other taxa) and the betweenness centrality (the number of times the taxa are passed in the shortest path between all nodes) of fungi were not higher than that of prokaryotes, they dominated the top 20th percentile (indicated by orange crosses in Fig. 3D). We then carried out keystone species analysis to identify taxa that represent the top 20th percentile of degree and betweenness. The keystones mainly were chemolithoautotrophic prokaryotes: methanotrophic and methanogenic archaea (ANME-1b, Marine Benthic Group D, and JS1), sulfate-reducing bacteria (SEEP-SRB1, *Desulfatiglans*, *Sulfurovum*, and *Hydrogenophilus*), and potential nitrite respirers (*Aminicentales*). The findings were justified considering the ability of chemolithoautotrophs to metabolize methane, sulfates, and nitrates into dissolved organic matter and support the higher trophic-level community (67), making them the center of biological activity. Roughly one-third of the

keystones were fungi ($n$ = 8), with three species (*Fusarium oxysporum*, *Suillus lakei*, and *Malassezia* sp.) coinciding with our previous observation of taxa that were distinct from seeping sediment (Fig. 2B). Most notable to us, fungi from the genus *Fusarium* were keystone taxa. *Fusarium solani* has been experimentally proven to solubilize and degrade methane (68), which further solidifies the potential role of fungi in mediating methane biogeochemistry. Overall, we establish statistically founded evidence that fungi play central and highly connected keystone roles in enhancing cold seep prokaryotic network robustness and efficiency.

**Fungal ASVs correlated with high methane activity play equally central roles as prokaryotic chemolithoautotrophs.** We then found fungal ASVs that correlate with methane using a selection of balances (selbal) with 2,000 iterations and 5-fold cross-validation at each step (total of 10,000 validation steps). Selbal is driven by a log-contrast model that addresses the compositionality of our microbiome data (69). The methane concentration across all cold seep samples followed a nonnormal distribution (Fig. S7); hence, traditional linear models can lead to spurious results. To tackle the issue, we discretized methane concentration into "high" and "low" relative to the surface layer (Fig. 1C). The division is driven by the belief that the SMTZ (10 to 12 cm) with the highest methane concentration and methanotrophic activity is the hub of biogeochemistry in cold seep sediment (here referred to as "high"). In contrast, the surface and bottom layers with lower methane concentration can be labeled as areas of lower methane-related activity (here referred to as "low"). Fungal ASVs that appeared at minimum in 30% of validation steps in favor of high methane concentrations were noted as "high-methane-signature fungi."

Five fungal ASVs favored high methane activity, two of which were left unidentified at the phylum level. *Fusarium oxysporum* and *Moniliella mellis* were two of the five ASVs classified with high confidence to the species level (>98% confidence). We first encountered two ASVs for favoring seep sediment (Fig. 2B), then for playing keystone roles in fungal-prokaryotic networks (Fig. 3C), and then for their correlation with areas of high methane concentration (notably the SMTZ) (Fig. 4B). Thus, we explored their role in our cold seep multidomain (BAF) network to elucidate specific interactions that might uncover their functional role in the microbial community (Fig. 4C). The interactive version of the BAF network can be accessed through our GitHub page (see "Data and code availability" below).

We filtered our network to include only taxa with three or more connected edges to focus on central ASVs of higher network-mediating relevance. The network was manually inspected, and multiple sources of literature were surveyed to categorize and validate their metabolic roles. Those with special metabolic functions were labeled with their highest-resolution taxonomic information. The majority of central ASVs were methanotrophic archaea and chemolithoautotrophic bacteria. Fungi formed a hub with *Fusarium solani*, *Fusarium oxysporum*, *Suillus lakei*, and *Moniliella mellis* at the core. All methane-correlated fungi (high-methane-signature fungi [Fig. 4C]) had negative interactions with one another, suggesting intercompetition. They likewise had only positive associations with ANME archaea, consistent with culture-based findings that filamentous fungi solubilize methane for methanotrophic prokaryotes and work synergistically in metabolizing methane (36). The ability of *Fusarium* and *Moniliella* to solubilize methane and degrade oils in experimental settings has been confirmed (68, 70). Although we could not find direct evidence of *Suillus lakei* and its input to cold seep-specific biogeochemistry, it was the most dominant species, with three representative ASVs in the center of our network.

Overall, our cold seep network and its interactions are highly consistent with culture-based literature in numerous ways: first, that it was dominated by chemolithoautotrophs, which are ecologically crucial in mediating the cold seep ecosystems; second, that the interaction of SRB and ANME being positive is a well-established scientific fact validated through numerous culture experiments (71); and lastly, that the fungal species identified as significant contributors in our study have substantial literature support on their role as biogeochemical mediators of cold seep abiotic factors such as methane.

**Scavenging a global data set of cold seep metagenomes and metatranscriptomes for fungal genes of interest.** As a final step in validating the functional role of fungi in cold seep sediment, we gathered and explored a global data set of 10 seep sites and their associated metagenomic and metatranscriptomic sequences, with the

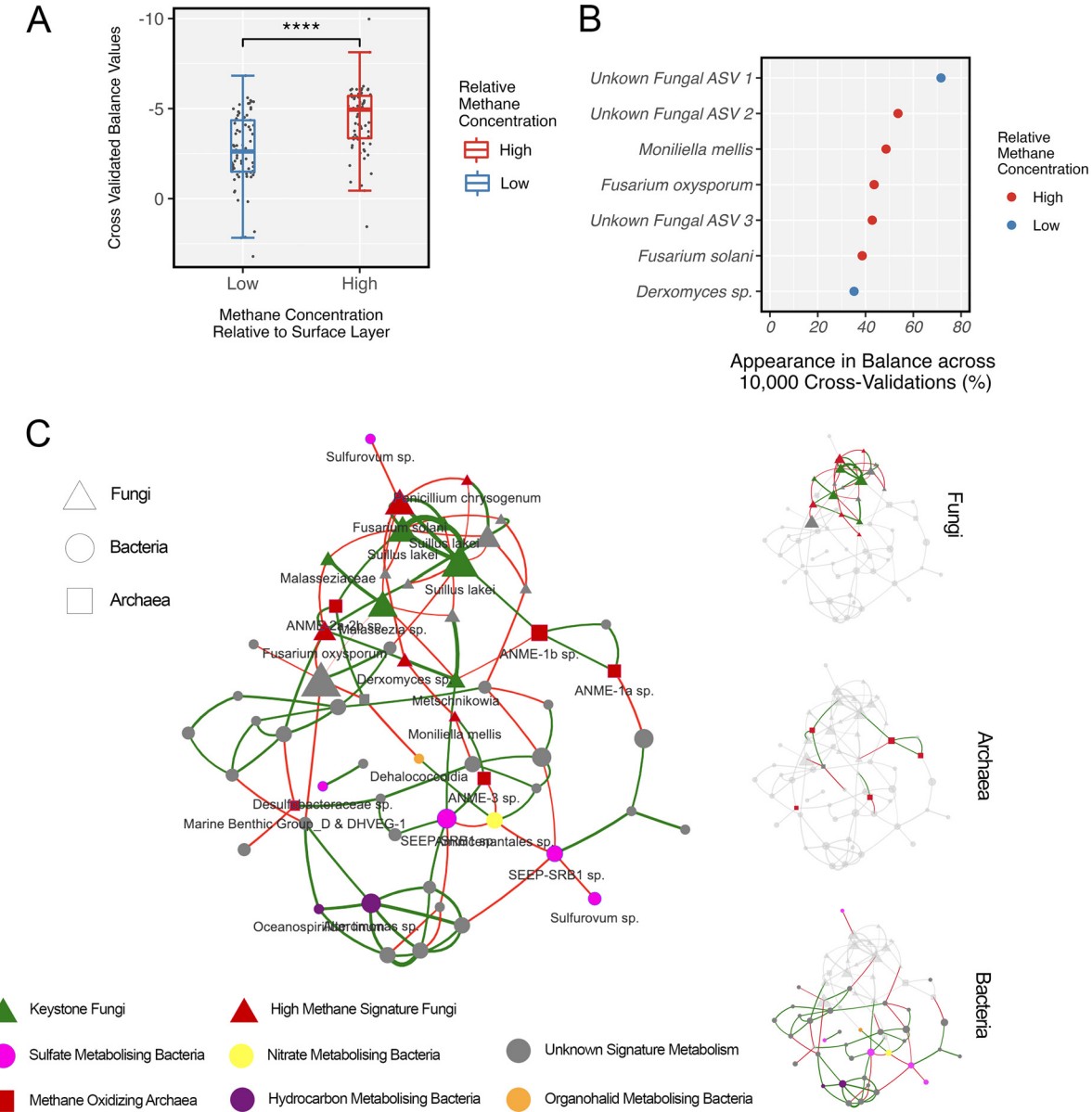

**FIG 4** Balance-driven correlation analysis using selbal of fungal 18S ASVs with discretized methane concentration into "high" and "low" based on methane levels in the sulfate-methane transition zone (SMTZ). (A) The exact values of balance ratios identified through 10,000 cross-validation iterations and significant difference. ****, $P < 0.0001$. (B) Appearances of each fungal ASV in the balance throughout 10,000 iterations. (C) Interactive cold seep network used to examine fungal, bacterial, and archaeal relationships with manually curated annotation of metabolism. Nodes represent ASVs and edges represent pseudocorrelation coefficients (derived from penalized estimators as reported by SpiecEasi). Green edges represent positive relationships and red edges represent negative covariance. Taxa of notable metabolic association with cold seep abiotic factors are colored, and ASVs with unidentified metabolic functions are gray.

largest done within this study (from the Haima cold seep) (Fig. 5A). The data sets were collectively analyzed to identify fungal genes of interest that play potentially essential roles in the seep biome. A summary of the sites is presented in Table S1.

Metatranscriptomes of three cold seep sites (Jiaolong, Haima, and Gulf of Mexico) were assembled and BLAST searched against a custom database of fungal genes of interest (see "Data and code availability" below). We assumed these genes to be responsible for their dominant role in cold seep networks, namely, genes encoding hydrophobins, cytochrome P450s, and the ligninolytic family of enzymes (laccase, lignin peroxidase, manganese peroxidase, and versatile peroxidase). Overall, we found +2,500 unique BLAST alignments with >97% sequence similarity to the NCBI database of all fungal sequences, which had significantly lower alignment scores with nonfungal candidates. The ligninolytic

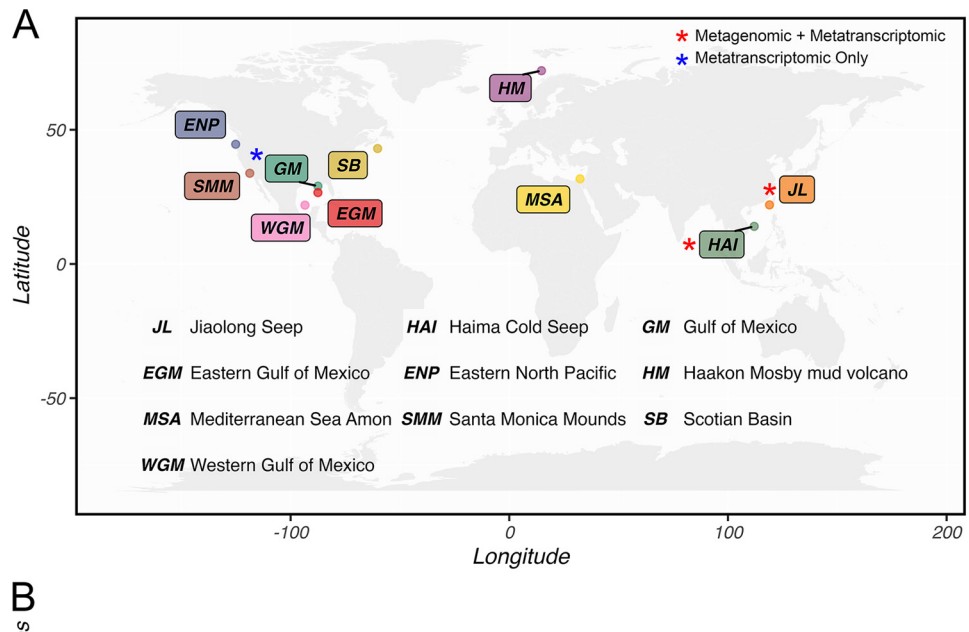

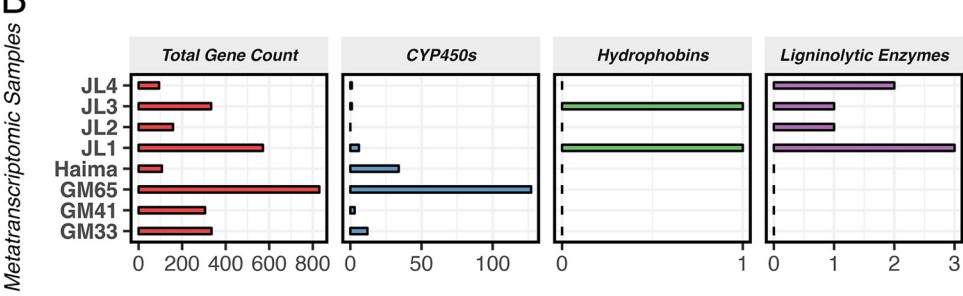

**FIG 5** (A) Overview of global metagenomic and metatranscriptomic samples used to scavenge fungal hydrophobin, cytochrome P450, and the ligninolytic family of enzymes. (B) Number of unique BLAST hits with more than 97% sequence identity against a custom data set of target genes in assembled metatranscriptomic data sets.

family of enzymes was found only in the Jiaolong site (Fig. 5B). We presume this to be due to the heterogeneity in the sediment composition of different regions. Fungal CYP450s had the most hits in the Gulf of Mexico and Haima seeps. Even though Jiaolong is considered a seep region, our previous studies show that it has considerably less methane bubbling than the Haima region (unpublished data). Further monitoring of methane activity and fungal CYP450 is needed to uncover its association with methane oxidation in the environment. Only two hydrophobin sequences were extracted (both from the Jiaolong region) and were partial sequences. Hydrophobins share very low sequence similarity to one another (as low as 10%) but have the same three-dimensional (3D) structure due to specific cysteine bonding patterns (72). A folding-aware homology search should be ideally performed to extract fungal hydrophobins from metagenomic sequences.

We chose a parallel metagenomic approach with a much larger data set than metatranscriptomes to identify our genes of interest but were unsuccessful. Out of 10 different pipelines attempted (Fig. S8), we found only the VEBA approach appropriate, which we discuss below. Overall, metatranscriptomics was proven to be the most effective method for monitoring fungal gene expression and activity.

## DISCUSSION

**Fungal-prokaryotic interactions and the role of fungi in cold seep microbial networks.** Fungi considerably increased the robustness of bacterial networks in the cold seep community. The random network attack demonstrated that BAF and BF networks were the most robust and could remain sizably intact despite removing large fractions of ecosystem members. *In situ*, this accidental removal could emulate any

stochastic elimination of a proportion of the community (e.g., as chemical contamination causing selective death of species, a seep metazoan invertebrate such as *Shinkaia crosnieri* consuming a proportion of the population, and sampling and removing sediment for our study). Fungi in this context have a broad set of supporting metabolites that help the growth and survival of neighboring bacteria, and it comes as no surprise that they enhance network connectivity and robustness. Filamentous fungi such as those seen in our fungal hub (Fig. 4C) can provide different structures colonizable by a large group of bacteria (73). At the same time, experimentally validated industrial models show improvement in phase transfer dynamics and bioavailability of methane to bacterial or archaeal methanotrophic associates (36).

Fungi likewise enhanced cold seep transformation efficiency in BF and BAF networks as evidence that they work to connect the members of the community and hence metabolite exchange. Saprotrophic fungi catalyze high-molecular-weight organic matter and polymers into particulate or dissolved form with extracellular enzymes in oxic and anoxic environments (74). This includes detritus produced by mussel beds, tubeworms, crabs, and in general, most observed meiofauna of cold seeps (Fig. 1) (75), which provide a redistribution loop of organic matter for carbon recycling through nutrient shortcuts. Fungal interactions with bacteria also shed light on secondary metabolite exchange and their impacts on quorum sensing (76), emphasizing the interactive role of fungi with other community members in enhancing connectivity and information exchange throughout the network.

Our results also show that fungi play dominant roles in BAF networks, larger than those of bacteria and archaea in terms of absolute edge weight (strength of association), and a group of fungi had the highest degree (connectivity) and betweenness (existing in the shortest path between most other organisms). Whether this is related to their metabolic complexity or heterogeneous community roles, insufficient scientific evidence exists to validate this hypothesis due to a lack of studies directly comparing the impacts of fungi and bacteria on community health. However, keystone species analysis showed that fungi, at the least, play a role equally as important as those of key chemolithoautotrophic bacteria and archaea. Previous studies on filamentous fungi showed that hydrophobins, a group of proteins on the surface of fungal hyphae, give them the ability to solubilize otherwise insoluble inorganic compounds such as methane and polycyclic aromatic molecules by increasing the partition coefficient, increasing their bioavailability by magnitudes (37, 77, 78). Previously, *Fusarium* spp. demonstrated an ability to solubilize and degrade methane (68). Alignment analysis done by Vergara-Fernández et al. (68) on the complete genome of the sexual form of *Fusarium solani* (*Nectria haematococca*) revealed the potential that homologs of the mmoC gene (methane monooxygenase component C) and the CYP63A2 gene (a versatile alkane monooxygenase P-450) could give insight into the methane oxidation and degradation abilities of the *Fusarium* genus, as well as explain its correlation with high methane activity in the SMTZ.

**Fungi as key downstream controllers of primary production in cold seeps.** Methanotrophs and other chemolithoautotrophic prokaryotes are integral to the ecosystem functioning in providing nutrients to the community at the base of the anaerobic food pyramid. By association, fungi are observed to control their metabolism by altering the bioavailability of their primary substrate can play equally essential roles in community-wide nutrient distribution. This raises the question of why scientists hesitate to pursue the role of fungi within biogeochemically essential areas such as cold seeps and what challenges we face when addressing the input of eukaryotes to ecosystems at large. Apart from filamentous fungi, other basal clades that cannot grow in mycelium forms (1) and their association with methanotrophic bacteria and archaea have not yet been subjected to study. A notable proportion of fungi (~40%) were unidentified even to the phylum level and had no close associates in the NCBI database, yet the unknown fungal ASVs were identified as key species within this study. This creates a bottleneck in further analysis and our understanding of the ecosystem caused by our lack of attempts to study microscopic basal fungi due to a historical bias in studying larger macrofungi with visible

vegetative structures such as Dikarya. Only recent ventures into protist-like fungi and unraveling their genomes have shown us their importance (79).

**Transcriptomics as the optimal tool for exploring the functional role of fungi in the deep sea.** While community composition analysis and network modeling can be an appropriate starting point for our study of fungi in isolated and niche environments like the cold seep, to ascertain the functional role of fungi, we must ultimately rely on omics data (metagenomics, metatranscriptomics, proteomics, and metabolomics). Our study is the first to report the active transcription of fungal genes in the seep environment. Previous studies relied on gene-level sequencing of the rRNA gene (either the internal transcribed spacer [ITS] or 18S), which does not directly assess the activity of eukaryotic organisms and cannot distinguish living and nonliving fungi. Methods such as RNA- and Propidium-monoazide (PMA)-based Illumina sequencing of eukaryotic amplicons are the key to overcoming our doubts about fungal activity in the region and can capture the living activity of microbial organisms (each with their shortcomings) (80). Our study used a metatranscriptomic data set not optimized for capturing eukaryotic genes (polyadenylation-based primers for RNA sequencing), and for this reason, we withheld from a thorough investigation of fungal transcript activity. It is also essential to know that eukaryotic organisms have a translational level of gene expression control, meaning that proteomics or metabolomics are needed to have a higher-level understanding of fungal function in the seep environment. Metagenomics as a tool was ultimately limited by the lack of appropriate *ab initio* genomic classification tools that do not rely on primary sequence alignment to a premade database. Microeukaryotes that live in isolated and niche marine environments like the cold seep have divergent sequences that do not resemble terrestrially sequenced models. This is further limited by the general fact that eukaryotic genomes are mainly composed of noncoding sequences and genes by introns. Overall, we believe RNA-based metatranscriptomic data to be the most appropriate way to move forward with our study of cold seep microeukaryotes.

**Importance and future of cold seep fungal research.** While many study cold seep bacteria, archaea, and (less so) fungi in isolation to uncover their independent importance, the true health of cold seep or any ecosystem relies on the interdependent nature of microcommunities. If methanotrophs provide energy for the ecosystem, fungi that control the availability of their substrate have equally important roles and vice versa. Hence, microbiomes should be studied as a whole to understand the true impact and value of the system. Another critical endeavor lies in disentangling the ability of fungi to metabolize and degrade methane. While sufficient evidence exists that some species of fungi can survive on methane (39, 68, 77), the exact mechanism and confirmation are up for discussion and require definite molecular and biochemical evidence. Comparative genomic studies have highlighted possible genes of interest, but no robust follow-up has been carried out (68). To enhance our understanding of fungi in a holistic sense, metaomics tools to infer complete and high-quality eukaryotic genomes exist and are being modified for compatibility with larger genomes (32, 81), providing valuable tools for studying nonculturable fungi. Last but not least, culturing endeavors can go hand in hand with metabolic analysis, as eukaryotic cells are proven to play important behavioral roles not discernible through omics data (61, 82).

**Conclusion.** In conclusion, our study highlights that fungi dominate cold seep sediment and enhance the robustness of microbial communities against random removal of taxa, provide shortcut paths to enhance the efficiency of information transfer, and play dominant roles side by side with chemolithoautotrophic methanotrophs and organohalide-respiring bacteria at the core of cold seep networks. The methane-solubilizing filamentous fungus *Fusarium* and oil-degrading yeast *Moniliella* were seen to form a fungal hub in the network core with mostly positive associations, supported by previous culture-based experiments. We are the first to report the active transcription of fungal genes in three cold seep sites across the world and have assessed both the metagenomic and metatranscriptomic approaches in analyzing fungal activity and report the latter to be most successful in assessing microeukaryotes in isolated biomes. Overall, we provide preliminary evidence for the dominating impact of fungi on cold seep microbial networks, as well as their role in enhancing the bioavailability of hydrophobic hydrocarbons to the community.

## MATERIALS AND METHODS

**Sample collection, DNA extraction, and porewater measurements.** Sediment samples ($n = 147$) were collected from methane seeps at water depths of ~1,400 m from the Haima region in the South China Sea during the HYDZ6-202102 cruise (R/V Haiyangdizhi VI, May 2021). Gravity cores at remotely operated regions were used to collect a surface 30-cm vertical profile of the target sediment. All samples were frozen immediately on board at −20°C, transferred to dry ice in the laboratory, and then stored in −80°C freezers until further use. Sediment subsections (as 5-cm vertical subdivisions of the 30-cm core) were taken from a total of four sampling regions: three cold seep sites (ROV1, ROV2, and ROV3) and one nonseep site (ROV5) that served as a control (Fig. 1). DNA was extracted from samples using the DNeasy PowerSoil Pro kit (Qiagen, USA) according to the manufacturer's protocol. Its quality was assessed by using the Qubit double-stranded DNA (dsDNA) assay kit with a Qubit 2.0 fluorometer (Life Technologies, USA) and checked by 1% agarose gel electrophoresis. In total, 147 samples (cold seep and control) were sent for 16S and 18S amplicon library preparation, and 21 purely cold seep sediments were sent for metagenomic analysis.

Environmental parameters relevant to the microbiome sediment samples were assessed in 50-cm vertical transects directly adjacent to the sampling layer. One core was used per ROV site with triplicate measurements. For each sample, the porewater was extracted and analyzed for chemical components by the Third Institute of Oceanography, Ministry of Natural Resources, in China. Concentrations of dissolved inorganic carbon (DIC) and isotopically labeled $^{13}$C of DIC were measured using a Delta V Advantage mass spectrometer (Thermo Scientific, USA) with a PoraPlotQ column (Agilent Technologies, USA) and reported in the standard conventional $\delta$ notation in per mille (‰) relative to the Vienna Pee Dee Belemnite (V-PDB). $CH_4$ was measured with an Agilent 6850 gas chromatograph (Agilent Technologies, USA), while sulfide ($S_2^-$) and phosphate ($PO_4^{3-}$) were measured using a Smartchem200 discrete autoanalyzer (Alliance, France). Sulfate ($SO_4^{2-}$) and ammonium ($NH_4^+$) cations and anion concentrations were examined using ion chromatography with an ISC-1000 instrument (Thermo Scientific, USA).

**Metaomics, 16S and 18S fungal DNA library creation, and sequencing.** For amplicon data, polymerase chain reaction (PCR) was used to amplify the appropriate rRNA gene barcode segments for community analysis, with an initial denaturation temperature of 95°C for 5 min, followed by 34 cycles of denaturation at 94°C for 1 min, elongation at 57°C for 45 s, and annealing at 72°C for 1 min, with a final extension at 72°C for 10 min. Prokaryotic 16S rRNA gene libraries were prepared by amplifying the V3-V4 region with forward primer 341F (5′-CCTAYGGGRBGCASCAG-3′) and reverse primer 806R (5′-GGACTACNNGGGTATCTAAT-3′) with an Illumina overhang adapter sequence using PCR to yield a "universal" representation of the prokaryotic community. Fungal 18S rRNA gene V4 region libraries were prepared similarly with the forward primer 528F (5′-GCGGTAATTCCAGCTCCAA-3′) and reverse primer 706R (5′-AATCCRAGAATTTCACCTCT-3′). PCR mixtures (25 $\mu$L) were prepared in triplicates (2.5 $\mu$L of 10-PCR buffer, 0.75 $\mu$L of 10 mM $MgCl_2$, 0.5 $\mu$L of 10 mM deoxynucleoside triphosphate [dNTP] mix, 0.5 $\mu$L of a 10 $\mu$M solution of primer, and 1 $\mu$L of Invitrogen Platinum *Taq* DNA polymerase [Life Technologies, USA]). The final PCR products were indexed following the instructions on Illumina's amplicon library preparation guide. The dsDNA concentration and the indexed amplicons' sizes were verified using the Qubit dsDNA high-sensitivity assay kit and a Qubit 2.0 fluorometer (Thermo Fisher Scientific, Canada) and the Agilent 2100 Bioanalyzer system (Agilent Technologies, Mississauga, ON, Canada), respectively. Indexed amplicons were pooled in equimolar amounts and pair-end sequenced using the NovaSeq 6000 PE250 platform (Illumina, USA). Data were further cleaned by removing adapter sequences, barcodes, poly-N-containing reads, and low-quality reads from the raw data.

For metagenomic samples, libraries were generated using the Illumina NEBNext Ultra DNA library prep kit (New England Biolabs [NEB], USA), sequenced with the NovaSeq6000 platform (Illumina, USA), and cleaned (150-bp paired-end reads) by removing barcodes, adapters, and poly-N-containing reads. The global omics data were retrieved from different publications and are readily accessible (Table S1) (83–91).

**Amplicon, metagenomic, and metatranscriptomic sequence data processing.** All amplicon data within this study were analyzed with the QIIME 2 pipeline version 2021.4 (92). Sequences were imported, demultiplexed, and trimmed for primer sequences. For 16S data, DADA2 was used to filter low-quality reads, denoise sequences, merge read pairs, and remove chimeras, followed by fine-level amplicon sequence variant (ASV) classification and counting in the form of an ASV count table. 18S sequences were processed similarly, with consideration of the high sequence length heterogeneity of the region. SILVA SSU Ref NR v138 (93) was used to train a naive Bayes classifier and assign taxonomic units to the 16S and 18S sequences. All poorly resolved classifications were confirmed with NCBI's BLAST. The 18S data set was filtered to include only fungal species. The output ASV tables, phylogenetic trees, and metadata were compiled in a Phyloseq R file (94) to facilitate data access and analysis reproducibility (see "Data and code availability" below).

For metagenomic and metatranscriptomic data, we used the VEBA v1.0.3 suite of tools (95) for its robust ability to use *ab initio*, non-database-reliant methods for binning eukaryotic genomes. All modules were used in default settings unless stated here. Metagenomes and metatranscriptomes were assembled with VEBA's megahit and rnaSPAdes, respectively. Three metatranscriptomic data sets were preassembled (84). Prokaryotic binning was performed (10 iterations) with the --skip-maxbin flag to remove eukaryotic contaminants, followed by microeukaryotic binning with the unbinned fasta sequences from the previous step. VEBA documents and logs of all the processes aforementioned for each module can be found on our Figshare page (see "Data and code availability" below). To identify fungal genes of interest from the metatranscriptome, we curated custom data sets using the NCBI nucleotide database (96) accessible through our data sharing link (see "Data and code availability" below).

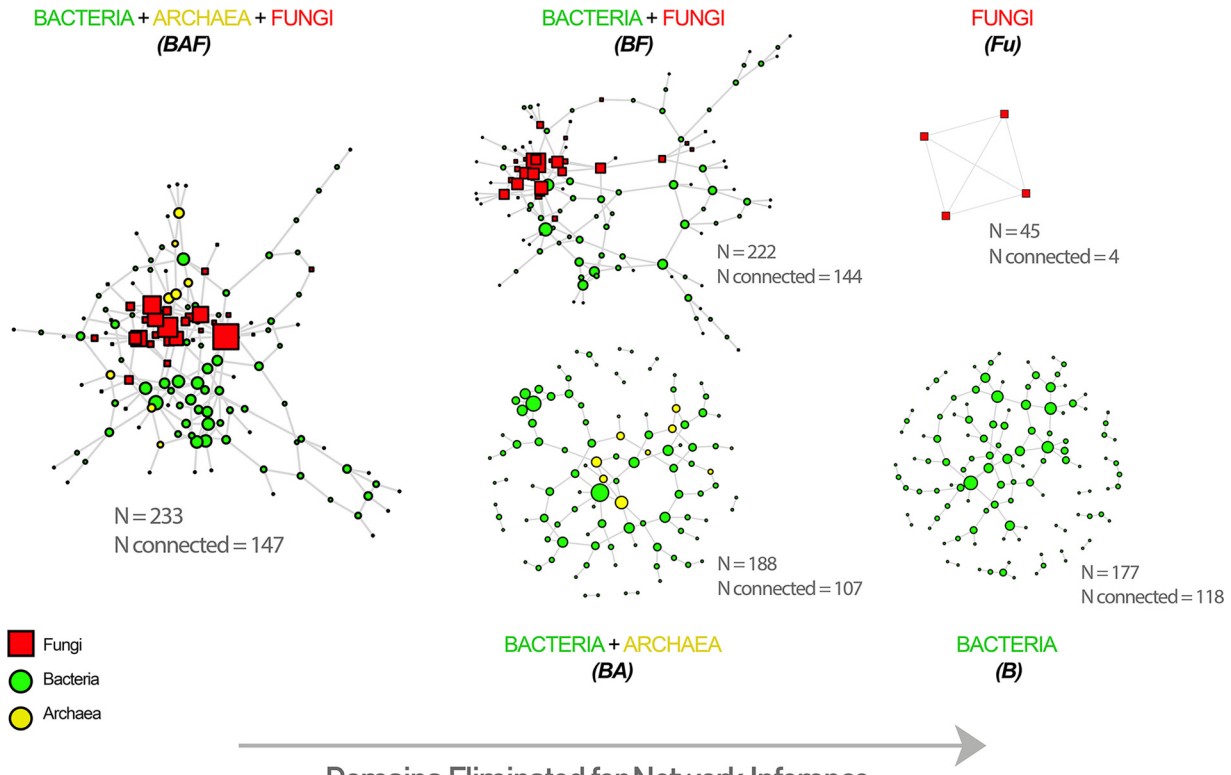

**FIG 6** Cold seep biological networks inferred within this study using taxa present in at least 20% of all samples (out of 130) in both ROV1 and ROV2 (most active seep sites). Original network was inferred using SpiecEasi SLR interdomain analysis with 16S and fungal 18S data represented by circle and square node shapes, respectively, resulting in bacterial-archaeal-fungal (BAF), bacterial-fungal (BF), bacterial-archaeal (BA), bacterial-only (Ba), and fungal-only (Fu) networks. Number of initial number of input taxa (N) and final connected components in the network (N connected) are likewise reported.

**Statistical analysis and phylogenetic trees.** For constructing 16S and 18S phylogenetic trees using sequence similarity in the QIIME2 environment, multiple alignment using fast Fourier transform (MAFFT) was performed with the q2-alignment plugin (97). Sequence masking was carried out to eliminate alignment columns that were phylogenetically uninformative or misleading, and phylogenetic trees were constructed using the approximately maximum likelihood approach with FastTree. Although 18S sequences alone are not suitable for generating accurate global-level taxonomic phylogenetic relationships, they were used as a visualization tool to present ASV-level fungal presence and relative abundance. All phylogenetic trees were visualized using the "ggtree" package on R (98). Compositional balance selection (selbal) (69) was performed with 5-fold cross-validation and 10,000 iterations. Differential abundance was performed using the compositional LinDA framework (59), and ASVs with adjusted $P$ values of less than 0.05 were used to identify precisely which group of taxa best separated the seep from control site samples.

All statistical analyses were performed using the R programming language version 4.11 with seeds to allow reproducibility when using pseudorandomized numerical generators. The R environment and all figures used in our study can be reproduced using our GitHub page with the "renv" environment manager (see "Data and code availability" below).

**Cold seep 16S-18S fungal network inference, analysis, and visualization.** All cold seep community samples were used for network inference ($n = 130$). Before network construction, ASVs present in less than 30% of samples or with a relative abundance lower than 0.0001 were removed from both 16S and fungal 18S data ($n_{16S} = 188$; $n_{fungal18S} = 45$) to retain focus on widely present interactions of community members and their relative impact on network properties. Networks were generated using the SpiecEasi software (to account for the compositional and zero-inflated nature of microbiome data to reduce false-positive detection rates) (62). The sparse and low rank (SLR) method was implemented to account for and remove the statistical effect of latent variables (e.g., sequencing depth and environmental parameters to reveal purely biological interactions [63]). Both 16S and fungal 18S data were used for cross-domain interaction analysis (nlambda = 20, minimum lambda ratio = 0.005, pulsar threshold = 0.05, and rep number = 20). The networks were created independently, and the $\beta$ parameter, which determines the impact of latent variables on the covariance matrix, was optimized using iterative network creation and model assessment using the extended Bayesian information criterion (99). The final connected components of the networks can be seen in Fig. 6.

The structural properties of networks (i.e., degree, centrality, and distance) were calculated using the igraph R package and documented equations (100). For pairwise statistical significance comparison of the degree, absolute weight, and betweenness centrality of bacteria, fungi, and archaea, the Wilcoxon signed-rank test was used, and $P$ values were inferred. Nodal information transfer efficiencies were

calculated using the definition provided by Latora and Marchiori (101) and compared using Welch's $t$ test after validating underlying assumptions of data suitable for the test (e.g., normality, outliers, and variances). Keystone species analysis was performed as described by Roume et al. (102), according to degree and betweenness centrality measures with cutoff values of the top 20th percentiles. The "core" network was obtained by filtering taxa according to the top 50th percentile of both degree and betweenness and subsequently visualized interactively with visNetwork to generate HTML links for interactive viewing. Species metabolic modes were classified using relevant literature with validated experiments.

**Network random-attack robustness.** Random network attack analysis followed the method of Iyer et al. (64) for node removal and its effects on the largest component connectivity (47), but a customized $R$ package was implemented to include iterative runs ($N = 10,000$). Normalized robustness ($R$) of all networks was calculated using the formula $R = \frac{1}{N}\sum_{i=1}^{N}\sigma\left(\frac{i}{N}\right)$ where $N$ is the initial size of the network, $\sigma$ is the relative size (to the original network size [$N$]) of the largest network component after node removal, and $i$ is the number of vertex or vertices removed from the network (i.e., the fraction of vertices removed relative to $N$). The vulnerability ($V$) of the networks was derived from robustness using the formula $V = 0.5 - R$. Welch's $t$ test was performed on the permutated vulnerability of the networks following a normal distribution, and $P$ values were reported.

**Data availability.** All raw sequencing reads done within this project have been uploaded to the SRA under projection accession number PRJNA849592. Phyloseq files for 18S and 16S amplicon data (including ASV tables, trees, taxonomy, and metadata) and metagenomic and transcriptomic assemblies, fungal gene annotations, and statistics can be found on Figshare (https://figshare.com/account/home#/projects/154346). The raw accession codes for outsourced metatranscriptomic and metagenomic reads are in the supplemental material. All customized code used to generate data can be accessed via the GitHub repository (https://github.com/erfanshekarriz/SeepFungiNet2022) and is readily reproducible with final and supplementary figures as outputs. Additional files that we have not listed are fully available upon request (https://orcid.org/0000-0002-1817-6050).

## SUPPLEMENTAL MATERIAL

Supplemental material is available online only.

**SUPPLEMENTAL FILE 1**, PDF file, 1.4 MB.

## ACKNOWLEDGMENTS

This study was supported by the Key Special Project for Introduced Talents Team of Southern Marine Science and Engineering Guangdong Laboratory (Guangzhou) (GML2019ZD0409) and the Hong Kong Branch of Southern Marine Science and Engineering Guangdong Laboratory (Guangzhou) (SMSEGL20SC01).

We declare that we have no competing interests.

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
