## [Reviewer comments · Microbiology Spectrum]

Microbiology Spectrum

Disentangling the Functional Role of Fungi in Cold Seep Sediment.

Erfan Shekarriz, Jiawei CHEN, Zhimeng XU, and Hongbin Liu

Corresponding Author(s): Hongbin Liu, The Hong Kong University of Science and Technology

Review Timeline:

Submission Date:	May 27, 2022
Editorial Decision:	October 17, 2022
Revision Received:	December 7, 2022
Editorial Decision:	December 21, 2022
Revision Received:	December 21, 2022
Accepted:	December 22, 2022

Editor: Noha Youssef

Reviewer(s): The reviewers have opted to remain anonymous.

Transaction Report:

DOI: <https://doi.org/10.1128/spectrum.01978-22>

October 17, 2022

Dr. Hongbin Liu
Hong Kong University of Science and Technology
Division of Life Science
Clear Water Bay, kowloon
Hong Kong
Hong Kong

Re: Spectrum01978-22 (Insights into the Central Roles of Fungi in Cold Seep Sediment as Revealed by Network Analysis)

Dear Dr. Hongbin Liu:

Thank you for submitting your manuscript to Microbiology Spectrum. I apologize for the length of time it took to secure reviewers that are experts in the field. Your manuscript has now been reviewed by one expert and their comments are appended to this email. When submitting the revised version of your paper, please provide (1) point-by-point responses to the issues raised by the reviewers as file type "Response to Reviewers," not in your cover letter, and (2) a PDF file that indicates the changes from the original submission (by highlighting or underlining the changes) as file type "Marked Up Manuscript - For Review Only". Please use this link to submit your revised manuscript - we strongly recommend that you submit your paper within the next 60 days or reach out to me. Detailed instructions on submitting your revised paper are below.

Link Not Available

Sincerely,

Noha Youssef

Journals Department
Reviewer comments:

Reviewer #1 (Comments for the Author):

Manuscript:

Spectrum01978-22
Insights into the Central Roles of Fungi in Cold Seep Sediment as Revealed by Network Analysis.

Summary:

The manuscript describes the sequencing and analysis of cold seeps microbiome, with a particular focus on the inference and interpretation of multi-domain ecological networks.

Deep sea sediment samples were collected from the South China Sea, DNA extracted and 16S/ITS regions were sequenced following PCR amplification.

The authors used a well-established data processing pipeline, qiime2, to turn the metagenomic data into organismal (ASV) relative abundances.

Secondly, the authors applied the SPIEC-EASI software package to infer association networks. Statistical properties of the static networks, as well as simulation experiments were used to assess network (and thus ecological) properties. The author's primary hypothesis was that fungi ASVs should have a special position in the multi-sample site network, based on known functional and metabolic interactions, and especially for methane oxidation in cold seep communities.

Consistent with these expectations, as well as from previous studies in other environments, the authors found that fungi are keystone species (network hubs) and their presence help to stabilize the inferred networks.

Major Comments:

The authors lay out a compelling case for cold seeps microbial communities being particularly interesting and, until now, under-studied ecosystems.

Additionally, while similar microbiome studies analyze OTU/ASV proportions and phylogeny (diversity) only, the authors apply a robust statistical inference procedure to study possible 'second-order' effects and present evidence for that strongly signals that particular fungi may play the key role in cold seep ecosystems.

The authors manage to squeeze quite a lot of insight out of what must be the first cold seep microbial ecological network.

However, I think there are several limitations in the study design and analysis that might limit its scientific impact.

1) The sample size is quite small, with $n=18$ from 3 sample sites. It is unclear, therefore, whether there is enough statistical power to truly limit false positives or if the communities are sampled enough to get enough to get a complete community census.

The later is especially since sequencing depth, species richness, rarefaction curves, etc are not reported in the manuscript.

On the other hand, the authors do address the former limitation by aggressively filtering ASVs prior to applying SPIEC-EASI.

An alternative, but related, tactic would be to aggregate ASV counts up to some fixed taxonomic level (e.g. genus).

2) Do the correlated ASVs occupy all (or even most) of the sampling sites?

If not, I struggle to understand if an inferred statistical association between ASVs between different sites could be meaningfully interpreted as a metabolic or other functional interaction.

If there are examples from the literature (even from author ecosystems), especially with in vitro/ co-culturing confirmatory evidence, credence would be added the underlying idea.

3) Environmental or other meta-data does not seemed to be fully leveraged in the microbiome analysis, despite that features such as chemical composition and DIC were apparently measured (L 93-). I especially hope that in-soluble methane data is available to the authors.

I understand that the SPIEC-EASI SLR method purports to adjust for these 'latent' environmental effects, but I recommend that the authors include at least a few ordination plots using the robust PCA approach detailed in that

preprint (or some other method) be added to this analysis.
For instance, the authors could correlate methane abundance with the scores of a robust PCA with and without the methane-degrading fungi included in the network. This would provide additional, indirect evidence of their functional role since, one might expect positive correlations with PCA scores when the methane degraders are included and ~0 correlation in their absence.
Proxy variables, such as sample site, could serve this goal as well.

4) The lack of functional data (metagenomic abundances, metabolomics, etc) unfortunately limits any stronger conclusions about the functional role fungi might play in the network.
Understandably, if only 16S/ITS data was available, and this limitation was already broached in the manuscript (e.g. L63).
However, any the assignment of functional roles to specific fungal ASVs is still overly qualitative.

One idea is that the authors could assess each [known] fungal ASV independently for it's ability to degrade methane (based on literature or a database search) *prior* to measuring their network centrality/betweenness.
Then the relationship between the ASV's keystone-ness and a putative functional role (e.g. ability to degrade methane) could be given as a quantitative score, e.g. a rank correlation or some other statistic.
Comparing these scores over relevant and irrelevant metabolic functions would give greater confidence to the 'methane hypothesis' and not require any new measurements.

Specific comments:

L143: "version 1.4.171" almost certainly refers to the version of RStudio, rather than the underlying R version. Both version should be reported.
L145: the package used to calculate diversity should also be cited.
It seems to be `vegan`.

L147/Fig 2C.

The authors should expand a bit on the formula choices for the regression model.
Was it simple visual inspection of the curve or was other model selection done?

Furthermore, I would correct the term "standard linearized regression".
"Standard linear regression" is probably what the authors meant, but this is a seems like an atypical name and I'm not coming up with many references for it - and I would still take to mean as containing only linear terms among the dependent variables.
Since there is only 1 single dependent variable here, the authors probably mean "simple linear regression", but since a polynomial model was also used they might want to say "simple and polynomial regression".

L156: The default parameter for the number of latent variables in SPIEC-EASI SLR ($h=2$) seems to have been used.
However, the SPIEC-EASI authors do extensive model selection in the preprint to find the optimal value of h , on a per dataset basis.
The choice of using the default vs empirically selecting the parameter should be at least discussed (but preferably, model selection via the eBIC criteria should just be done).

L294: 'parabolic' rather than 'hyperbolic' is more technically correct in describing the shape of a one-piece U-shape curve

Fig 2 A/B

The resolution of parts A/B the figure is quite low - I could not read the text with genus species names.

The legend is labelled Log-normalized relative abundance, but the scale is between 0 and 1, so this cannot be true. I looked for the code for guidance, but the scripts to reproduce the ggtree plots seem to not be available.

Fig 2 C:

The abbreviation (m) is not defined. Assuming it is meters?

Fig 3:

The networks are quite small here, so it's probably fine to remove isolated nodes from the plot and report those numbers in the figure legend.

Code:

A README with a short usage/dependent package installation guide would be useful.

I was able to run the scripts, but only after modifying hard-coded paths
- the phyloseq RData files seem to be coded relative to the authors home directory, rather than contained in the repo.

The interactive graph is quite nice!

Staff Comments:

Preparing Revision Guidelines

Please return the manuscript within 60 days; if you cannot complete the modification within this time period, please contact me. If you do not wish to modify the manuscript and prefer to submit it to another journal, please notify me of your decision immediately so that the manuscript may be formally withdrawn from consideration by Microbiology Spectrum.

Response to Reviewers

Microbiology Spectrum

October 28th, 2022

Dear respected Reviewers,

First and foremost we thank you for your time in revising this manuscript and providing critical feedback and suggestions that we personally consider very valuable in enhancing the quality and integrity of our work. We are also happy to inform you that nearly all your major and minor comments have been addressed, elevating the potential impact and rigor of our study. The principal modifications were the following:

- Firstly, we have **enhanced the size of our dataset by manifolds with the 18S Fungal primer (from $n=18$ to $n=130$ cold seep sites)**. This new dataset **includes a new seep site (ROV1)** that has the highest level of methane bubbling, methane concentration, and megafauna diversity than other regions, making it a great case study for studying cold seep fungal biodiversity.
- Secondly, all the new local samples have been analyzed for over **10+ nutrient environmental data including insoluble methane, dC13 methane, sulfate, nitrates, stable carbon isotopes**, and other factors that we believe to be vital in elucidating and establishing the role of fungi in cold seep nutrient cycles. The environmental parameters have also been leveraged to enhance our understanding of fungal input in cold seep biogeochemistry.
- Next, **our code has been cleaned up and made with a renv R studio lock file that allows full reproducibility (with a step-by-step README.md file)**. All server-based and non-R programs have been conducted using VEBA v1.0.3, which have log files with all information needed for reproducibility (deposited in a Figshare link). Three colleagues tested the code and found the amplicon/network analysis to be understandable and reproducible in less than one hour.
- Lastly, we **sequenced and analyzed our local dataset's metagenomic samples ($n=21$) and combined it with a global dataset of 10 cold seep metagenomic sites and 3 metatranscriptomic sites**. We found fungi to be actively expressing genes in the cold seep sediment using metatranscriptomics, making us the first study to report the functional presence of fungi in the region.
 - We tested 10 different pipelines to extract fungal genes from metagenomes but were unsuccessful in most. This is due to the fact that eukaryotic genes are mostly composed of introns and non-coding sequences, and hence highly divergent in sequence. Almost all metagenomic strategies favor prokaryotic binning and annotation, relying on sequence alignment of pure exons. We still would like to report our failures to highlight the unfortunate lack of available and robust tools for annotating eukaryotic genomes from smaller metagenomic datasets.
 - The metatranscriptomics approach was significantly more successful as it overhauls the aforementioned challenges, despite the dataset being significantly smaller. We were able to identify fungal genes of interest and report them.

With all changes being made, we are also excited to report that our novel results are **highly consistent** with the previous. This is in spite of the fact that the datasets have some minor but key differences:

1. The new dataset has a completely new cold seep site (ROV1)
2. The new dataset uses the 18S gene rather than ITS (although the taxonomic resolution was checked to be consistent with ITS)
3. The new dataset examines the surface layer (0-30cm at 5cm intervals) rather than the 8m depth. We chose surface layer samples to focus on the Sulfate Methane Transition Zone (SMTZ), the most important layer for cold seep methane biogeochemistry (Hu et al., 2017).

For the reason that we have found exciting new findings, enhanced the robustness of our statistics, and found direct correlations between fungi and metadata, we have chosen to rewrite our manuscript with higher confidence and certainty. Our conclusions remain the same as our first draft, but the language and tone have been synchronized with our newfound level of analytic robustness.

Overall, to the best of our knowledge, our dataset is the largest cold seep study encompassing multiple nearby sites sequencing cross domains (Bacteria, Archaea, and Fungi) and analyzing a multitude of environmental parameters (methane, sulfate, nitrate, etc.). Our study also leverages a diverse set of omics approaches and a global cold seep metagenomic dataset to bring forth the role of fungi in cold seeps in spite of the challenges that we face in analyzing eukaryotic metagenomic data.

Being the first publication that uses appropriate compositional statistical models to infer the functional and important role of fungi in cold seep sediment, we feel confident in the contribution of our work to the scientific community and hope to encourage others in using multi-omics methods to uncover the vital role of eukaryotes in challenging environments.

We thank you for your consideration and highly useful comments in making our publication more robust, confident, and well-presented.

Erfan Shekarriz,
Master of Philosophy Researcher
Department of Ocean Science
Hong Kong University of Science & Technology

COMMENT BY COMMENT RESPONSE

- **Major Comments**

The authors lay out a compelling case for cold seeps microbial communities being particularly interesting and, until now, under-studied ecosystems. Additionally, while similar microbiome studies analyze OTU/ASV proportions and phylogeny (diversity) only, the authors apply a robust statistical inference procedure to study possible 'second-order' effects and present evidence that strongly signals that particular fungi may play the key role in cold seep ecosystems. The authors manage to squeeze quite a lot of insight out of what must be the first cold-seep microbial ecological network.

However, I think there are several limitations in the study design and analysis that might limit its scientific impact.

1) The sample size is quite small, with $n=18$ from 3 sample sites. It is unclear, therefore, whether there is enough statistical power to truly limit false positives or if the communities are sampled enough to get enough to get a complete community census. The later is especially since sequencing depth, species richness, rarefaction curves, etc are not reported in the manuscript. On the other hand, the authors do address the former limitation by aggressively filtering ASVs prior to applying SPIEC-EASI. An alternative, but related, tactic would be to aggregate ASV counts up to some fixed taxonomic level (e.g. genus).

- The sample size has been enlarged to $n=130$ in three active regions of the Haima cold seep, and $n=17$ samples in control (one more active seep site added).
- The sample-based rarefaction curves were checked and all samples attained maximum coverage (supplementary fig 1)
- The ROV sample site-based rarefaction curves were reported as a measure of alpha diversity in Fig 2.
- The aggressive filtering was still performed to ascertain that associations inferred were highly confident

2) Do the correlated ASVs occupy all (or even most) of the sampling sites? If not, I struggle to understand if an inferred statistical association between ASVs between different sites could be meaningfully interpreted as a metabolic or other functional interaction. If there are examples from the literature (even from author ecosystems), especially with in vitro/ co-culturing confirmatory evidence, credence would be added the underlying idea.

- A piece of code called (7_filternetwork.R) has been implemented to remove ASVs that were not found in both ROV1 and ROV2. These two sites have the highest concentration of methane. Only 5 ASVs were removed.
- The network resulting from our analysis has clear positive connections between Anaerobic Methanotrophic Archaea (ANME) and Sulfate Reducing Bacteria (SRB), which have been extensively studied in the field of cold seep microbiology (Cui et al., 2015). The primary producers also had mostly negative associations with one another suggesting competition for resources. The network was dominated by chemolithoautotrophs, which we also consider as evidence of reliability.

3) Environmental or other meta-data does not seem to be fully leveraged in the microbiome analysis, despite that features such as chemical composition and DIC were apparently measured (L 93-). I especially hope that in/soluble methane data is available to the authors. I understand that the SPIEC-EASI SLR method purports to adjust for these 'latent' environmental effects, but I recommend that the authors include at least a few ordination plots using the robust PCA approach detailed in that preprint (or some other method) be added to this analysis. For instance, the authors could correlate methane abundance with the scores of a robust PCA with and without the methane-degrading fungi included in the network. This would provide additional, indirect evidence of their functional role since, one might expect positive correlations with PCA scores when the methane degraders are included and ~ 0 correlation in their absence. Proxy variables, such as sample site, could serve this goal as well.

- The metadata was described as evidence and confirmation of cold seep activity.
- Correlation analysis was performed between sample metadata and different aspect of fungal microbial ecology such as function, diversity, etc.
- We've examined the robust PCA approach mentioned above. The scree plot of the PCA analysis shows that there is no specific principal coordinate that can efficiently summaries our high dimensional data, which we feel would not be appropriate for correlation with methane. Readers can still navigate our results in the (17_robustPCA.R) file and explore the data as they wish. A correlation heatmap of the robust PCs of different networks with the environmental parameters can also be generated using this file.

4) The lack of functional data (metagenomic abundances, metabolomics, etc) unfortunately limits any stronger conclusions about the functional role fungi might play in the network. Understandably, if only 16S/ITS data was available, and this limitation was already broached in the manuscript (e.g. L63). However, any the assignment of functional roles to specific fungal ASVs is still overly qualitative.

One idea is that the authors could assess each [known] fungal ASV independently for it's ability to degrade methane (based on literature or a database search) *prior* to measuring their network centrality/betweenness. Then the relationship between the ASV's keystone-ness and a putative functional role (e.g. ability to degrade methane) could be given as a quantitative score, e.g. a rank correlation or some other statistic. Comparing these scores over relevant and irrelevant metabolic functions would give greater confidence to the 'methane hypothesis' and not require any new measurements.

- Functional metagenomic & metatranscriptomic analysis has been reported!

- **Specific comments:**

L143: "version 1.4.171" almost certainly refers to the version of RStudio, rather than the underlying R version. Both version should be reported. L145: the package used to calculate diversity should also be cited. It seems to be `vegan`.

- The typo that has been corrected
- Renv lockfile has all packages used and the appropriate version

L147/Fig 2C.

The authors should expand a bit on the formula choices for the regression model. Was it simple visual inspection of the curve or was other model selection done? Furthermore, I would correct the term "standard linearized regression". "Standard linear regression" is probably what the authors meant, but this is a seems like an atypical name and I'm not coming up with many references for it - and I would still take to mean as containing only linear terms among the dependent variables. Since there is only 1 single dependent variable here, the authors probably mean "simple linear regression", but since a polynomial model was also used they might want to say "simple and polynomial regression".

- This analysis has been removed. The typo for standard linear regression has been resolved.

L156: The default parameter for the number of latent variables in SPIEC-EASI SLR ($h=2$) seems to have been used. However, the SPIEC-EASI authors do extensive model selection in the preprint to find the optimal value of h , on a per dataset basis. The choice of using the default vs empirically selecting the parameter should be at least discussed (but preferably, model selection via the eBIC criteria should just be done).

- This is something that we originally completely missed when reading the preprint. Thank you for bringing our attention to it. Model selection for the β parameter has been performed.
- The parameter β or rank (r) has been thoroughly tuned using the method described in the original model and evaluation using eBIC (see code `spieceasi_SLR.R`)

Specifically, the preprint authors use a stepwise exponential ascend:

```
ranks <- round(exp(seq(log(2), log(32), len=6)))
se.slr <- spiec.easi(phy_taxfilt, method='slr', nlambda=50,
  lambda.min.ratio=1e-2, r=ranks, lambda.log=TRUE,
  pulsar.params=list(ncores=32, rep.num=30))
se.slr$ebic <- sapply(se.slr, function(x)
  ebic(x$refit$stars, x$est$data,
    x$est$loglik[x$select$stars$opt.index]))
```

- Each network (BAF, BA, BF, B, Fu) has been now constructed independently as we assume different domain interactions are impacted differently to varying extents by the environmental/latent variables. The β has also been independently tuned.

L294: 'parabolic' rather than 'hyperbolic' is more technically correct in describing the shape of a one-piece U-shape curve

- The analysis has been redacted.

Fig 2 A/B

The resolution of parts A/B the figure is quite low - I could not read the text with genus species names. The legend is labelled Log-normalized relative abundance, but the scale is between 0 and 1, so this cannot be true. I looked for the code for guidance, but the scripts to reproduce the ggtree plots seem to not be available.

- The ITS-level phylogenetic tree has been relocated to the supplementary material and legend units have been fixed.
- The scripts for all figures are available and labeled clearly to allow full reproduction.

Fig 2 C:

The abbreviation (m) is not defined. Assuming it is meters?

- The abbreviation is now defined

Fig 3:

The networks are quite small here, so it's probably fine to remove isolated nodes from the plot and report those numbers in the figure legend.

- Networks enlarged and isolated nodes reported independently.

Code:

A README with a short usage/dependent package installation guide would be useful. I was able to run the scripts, but only after modifying hard-coded paths - the phyloseq RData files seem to be coded relative to the author's home directory, rather than contained in the repo.

- README file has been made with clear instructions to reproduce the data
- All files have been coded relative to the R-project directory, which can be directly cloned from GitHub using the README instructions.

The interactive graph is quite nice!

- Interactive graph has been reconstructed!

December 21, 2022

Dr. Hongbin Liu
The Hong Kong University of Science and Technology
Department of Ocean Science
Clear Water Bay, kowloon
Hong Kong
Hong Kong

Re: Spectrum01978-22R1 (Disentangling the Functional Role of Fungi in Cold Seep Sediment.)

Dear Dr. Hongbin Liu:

Thank you so much for addressing the reviewer's comments. There is just one more issue to address about the LDA scores of Lefse. If you can do that and resubmit a revision, I would really appreciate it. Again, thanks for taking the reviewer's comments to heart and modifying your manuscript accordingly.

Link Not Available

Sincerely,

Noha Youssef

Journals Department
Reviewer comments:

Reviewer #1 (Comments for the Author):

Manuscript:

Spectrum01978-22
Disentangling the Functional Role of Fungi in Cold Seep Sediment.

I thank the authors for addressing all the major and minor concerns in this revision. I agree that the additional data adds to the scope and clarity of the aims.

The new manuscript more carefully implements the original methodology and, importantly, provides additional scientific impact. The role of fungi in the cold seep ecosystem are additionally highlighted with clearer network graphics that visually corroborate the 'random attack' analysis.

Additionally, the inclusion of greater sample sizes, more meta-data as well as transcriptomics data alleviate all of my original concerns.

I have only one specific comment regarding the use of LDA within the LEfSe. As far as I understand it, LDA step is carried out on relative abundance data are not compositionally robust (see Gloor et al doi:10.1016/j.annepidem.2016.03.003).

Since this was a major impetus to use SPIEC-EASI and selbal, can you add a comment on how you are dealing to compositional closure in the differential abundance analysis (or perhaps choose a different method?)

Staff Comments:

Preparing Revision Guidelines

Please return the manuscript within 60 days; if you cannot complete the modification within this time period, please contact me. If you do not wish to modify the manuscript and prefer to submit it to another journal, please notify me of your decision immediately so that the manuscript may be formally withdrawn from consideration by Microbiology Spectrum.

Response to Reviewers

Microbiology Spectrum

Dec 22nd, 2022

Dear respected Reviewers,

We thank you again for your time and consideration.

We initially performed LEfSe while being aware that it is not highly suitable for compositional data and can result in false-positives. We kept our analysis since we saw that it was supported by other data later in our findings, and did not place emphasis on it.

Despite the fact, we agree that if all analysis in the manuscript is done with compositionally aware techniques our findings and dedication to robust compositional analysis of microbiome data will be more coherent and consistent. Hence, we have made a switch to LinDA a linear model for differential abundance analysis of microbiome compositional data (<https://doi.org/10.1186/s13059-022-02655-5>).

Our findings that *Fusarium oxysporum* dominate seep sediment remains consistent. We do still highlight that differential abundance performed on field samples is not the most detrimental in our conclusions on the importance of fungi in cold seep sediment.

Best and thank you again,

Erfan Shekarriz,
Master of Philosophy Researcher
Department of Ocean Science
Hong Kong University of Science & Technology

COMMENT BY COMMENT RESPONSE

- **Minor Comments**

I have only one specific comment regarding the use of LDA within the LEfSe. As far as I understand it, LDA step is carried out on relative abundance data are not compositionally robust (see Gloor et al doi:10.1016/j.annepidem.2016.03.003). Since this was a major impetus to use SPIEC-EASI and selbal, can you add a comment on how you are dealing to compositional closure in the differential abundance analysis (or perhaps choose a different method?)

- The analysis has been replaced with LinDA., a linear model for differential abundance analysis of microbiome compositional data (<https://doi.org/10.1186/s13059-022-02655-5>)

December 22, 2022

Dr. Hongbin Liu
The Hong Kong University of Science and Technology
Department of Ocean Science
Clear Water Bay, kowloon
Hong Kong
Hong Kong

Re: Spectrum01978-22R2 (Disentangling the Functional Role of Fungi in Cold Seep Sediment.)

Dear Dr. Hongbin Liu:

Thank you for taking care of the Lefse issue.

Your manuscript has been accepted, and I am forwarding it to the ASM Journals Department for publication. You will be notified when your proofs are ready to be viewed.

Sincerely,

Noha Youssef
Editor, Microbiology Spectrum
